# Metadata-guided Consistency Learning for High Content Images

**Johan Fredin Haslum** [1,2,3]                                    JHASLUM@KTH.SE

**Christos Matsoukas** [1,2,3]                                     MATSOU@KTH.SE

**Karl-Johan Leuchowius** [3]             KARL-JOHAN.LEUCHOWIUS@ASTRAZENECA.COM

**Erik Müllers** [3]                            ERIK.MUELLERS@ASTRAZENECA.COM

**Kevin Smith** [1,2]                                              KSMITH@KTH.SE

[1] *KTH Royal Institute of Technology, Stockholm, Sweden*

[2] *Science for Life Laboratory, Stockholm, Sweden*

[3] *AstraZeneca, Gothenburg, Sweden*

**Editors:** Accepted for publication at MIDL 2023

## Abstract

High content imaging assays can capture rich phenotypic response data for large sets of compound treatments, aiding in the characterization and discovery of novel drugs. However, extracting representative features from high content images that can capture subtle nuances in phenotypes remains challenging. The lack of high-quality labels makes it difficult to achieve satisfactory results with supervised deep learning. Self-Supervised learning methods have shown great success on natural images, and offer an attractive alternative also to microscopy images. However, we find that self-supervised learning techniques underperform on high content imaging assays. One challenge is the undesirable domain shifts present in the data known as *batch effects*, which are caused by biological noise or uncontrolled experimental conditions. To this end, we introduce Cross-Domain Consistency Learning (CDCL), a self-supervised approach that is able to learn in the presence of batch effects. CDCL enforces the learning of biological similarities while disregarding undesirable batch-specific signals, leading to more useful and versatile representations. These features are organised according to their morphological changes and are more useful for downstream tasks – such as distinguishing treatments and mechanism of action.

**Keywords:** Representational Learning, Domain Shifts, High Content Screening

## 1. Introduction

High content screening (HCS), or image-based screening of cells treated with potential drug-molecules have become an important tool for pharmaceutical drug discovery and development. Image-based screens enable us to probe, at scale, large panels of compounds and other perturbents. These screens capture rich phenotypes of complex sub-cellular processes and tie them to a multitude of endpoints, such as mode of action, bioactivity and toxicity. Large scale screening efforts of standardized HCS assays such as Cell Painting (Bray et al., 2016) in the JUMP-CP initiative, have generated phenotypic response data for hundreds of thousands of treatments. However the extent to which, and how, this valuable information can be extracted from cell culture images using machine learning is still an open question.

For years, feature extraction methods like Cell Profiler (Carpenter et al., 2006) were used to automate analyses of HCS screens, but in recent years deep learning methods have

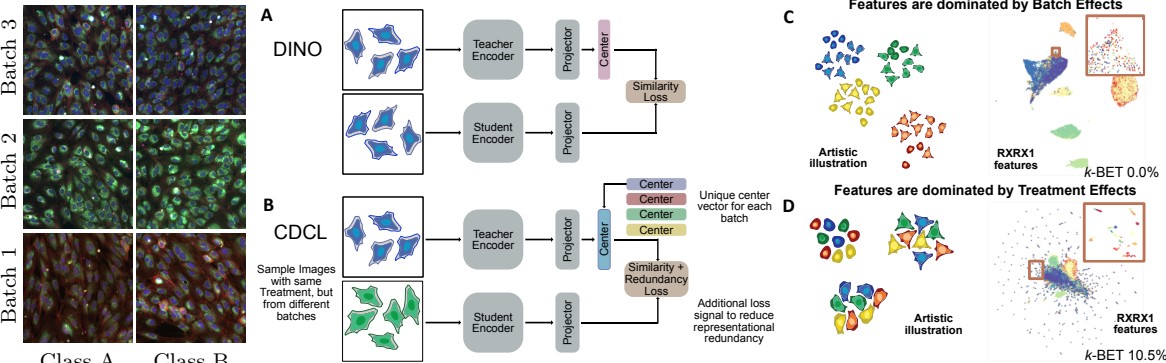

Figure 1: *Our approach.* In HCS data, each image (left) contains two dominating signals. The first is related to the treatment and its biological effect. The other is comprised of confounding factors, *i.e.* batch effects. (A) DINO results in features dominated by batch effects (C). (B) depict CDCL and the resulting feature embeddings (D) which correctly capture treatment effects and ignore batch effects. Right panels are colored by batch, zoomed area is colored by treatment.

proven more successful (Hofmarcher et al., 2019; Moshkov et al., 2022). However, applying supervised learning for HCS is challenging. The scale of the experiments mean the time and cost to collect labeled data is prohibitively expensive. Furthermore, the data are often noisy and tailored to specific experimental settings, limiting their usefulness for other tasks.

Self-Supervised Learning (SSL) seems to be an appealing alternative. SSL approaches use various techniques to circumvent the need for labeled samples, and can in the process mitigate the effects of noisy or erroneous annotations by simply not using any labels. Recent advances in SSL in natural imaging have been shown to generate more useful features than their supervised counterparts (Caron et al., 2021). Therefore, HCS would appear to be an excellent candidate application for self-supervised learning.

We find that, counterintuitively, self-supervised learning techniques do not exhibit the same success for HCS as they do for natural images. The reason for this failure is an issue at the heart of most HCS datasets – *batch effects*[1]. Batch effects are a well-known problem in HCS referring to undesirable domain shifts present in the data caused by biological noise or difficult-to-control experimental conditions (see Figure 1). They are confounding factors that can dominate the signal, *e.g.* minute variations in the fluorescent stains used or the length of incubation times can lead to important differences in the acquired images. These confounding factors are related to the batch (a group of similar experiments) the data was generated in – not the biological process of interest. As such, they are problematic for SSL, which tends to be misled into learning features dominated by batch-related factors. Is there a way for SSL to learn representations that are useful for HCS data?

We propose a fix to modern self-supervised learning methods that overcomes domain shifts/batch effects, called Cross-Domain Consistency Learning (CDCL). The main idea is to utilize knowledge of the domain (or batch) from which the data is sourced to enforce

---

1. Note that the term "batch" in the context of HCS refers to a group of experiments generated under similar conditions, as opposed to the samples presented to the model for learning at each iteration in the ML context. To avoid confusion we use the term "mini-batch" when we refer to the latter.

consistent representations across the domains/batches. We use readily available metadata of a sample (*i.e.* the batch ID or the treatment ID) during learning to choose examples from different domains/batches but with the same treatment, and force the representations to be similar using consistency learning. This forces the model to focus on the biological signal of interest while disregarding confounding factors related to batch effects. We demonstrate the effectiveness of the proposed approach in two fluorescence microscopy datasets and show that our method results in versatile and biologically meaningful features – better suited for down-stream analysis. Our contributions are summarised as follows:

- We showcase the limitations of current SSL methods in high content imaging attributed to batch effects.

- We propose CDCL, a method that counteracts batch effects and enables SSL to work for HCS data. This is accomplished through three technical innovations described in Section 3: (1) a metadata based sampling technique, (2) an additional loss signal, and (3) a domain-based normalization of the features.

- We demonstrate that the representations learned using our method are more relevant for downstream tasks.

These findings, along with additional studies, suggest that SSL-inspired methods can indeed be utilized for sparse and noisy high content screening data, opening the door to benefit from deep learning where it was previously difficult. The code to reproduce our work is available at https://github.com/cfredinh/CDCL.

## 2. Related Work

Feature extraction from high content imaging microscopy data has historically been done using software like Cell Profiler (Carpenter et al., 2006; Simm et al., 2018; Warchal et al., 2019; Trapotsi et al., 2022), relying on single cell segmentation followed by human-crafted feature extraction. More recent works have shown the benefits of deep learning-based feature extraction (Moshkov et al., 2022). These features have been used both directly from ImageNet trained weights (Moshkov et al., 2022) as well as models trained for the particular task at hand, using ground-truth labels or available metadata (Caicedo et al., 2018; Hofmarcher et al., 2019; Moshkov et al., 2022).

**Self-supervised learning** methods use various techniques to circumvent the need for labeled samples. SSL approaches have made significant improvements in recent years and are closing the performance gap to supervised learning on commonly used datasets such as ImageNet (Grill et al., 2020; Caron et al., 2021). At the core of the most recent SSL approaches is the use of consistency learning between positive pairs, enforcing similar output distributions (Chen and He, 2021). Contrary to previous similarity based approaches, these methods do not require negative examples, separating them from pure contrastive methods. This also means that they do not require large batches of negative samples or hard negative mining in order to balance the loss (Chen et al., 2020). They can be applied more easily and have been shown empirically to produce state-of-the-art results (Chen and He, 2021; Caron et al., 2021; Grill et al., 2020).

**Contrastive learning in high content imaging** – previous works have explored the use of contrastive learning based approaches in HCI, with metric learning approaches using contrastive losses being used both in supervised (Ando et al., 2017) as well as weakly supervised manners (Caicedo et al., 2018). This was followed by more recent work exploring contrastive self-supervision methods (Janssens et al., 2021; Perakis et al., 2021). Beyond this other methods such as Archetypal Analysis (Siegismund et al., 2021) have been explored for representational learning. However, all of these works primarily explore small datasets with few samples, treatment and task specific classes in a translucent setting, thus avoiding working on problems associated with batch effects. Further, none of the recent successful SSL methods that enforce consistency instead of contrastive losses have yet been explored. We consider newer, more powerful SSL methods in HCS datasets, focusing on their ability to learn good representations in larger and more challenging datasets.

## 3. Methods & Experimental Setup

We start by investigating the extent to which recent SSL-based methods can be utilized for representational learning in high content imaging datasets – what is the best way to learn useful representations for fluorescence microscopy data? We consider two primary tasks: a) cross-batch treatment identification and b) identifying the mechanism-of-action – a downstream task. Both tasks require useful and consistent representations. Below, we propose CDCL, a method to counteract the domain shifts caused by batch effects and learn useful representations. For comparison we introduce, as baselines, methods to learn representations using weakly-supervised learning of treatments as well as two recent state-of-the-art consistency-based SSL methods: BYOL and DINO. In Section 4, we show why current SSL methods fail to learn useful biological features.

### 3.1. Baselines

High content screens are rarely accompanied with large quantities of task-specific labels, which limits the applicability of supervised learning. This lack of labels can be attributed to the exploratory nature of these screens, making ground truth labels difficult to define for each treatment, especially at scale. However, metadata such as compound and concentration information is commonly available. Such data is therefore often used as "weak" labels (Caicedo et al., 2018). We consider each treatment as a unique class and assume that each treatment generates a different phenotypic response. In this work, we use the "weakly"-supervised approach as baseline, where the prediction of treatments is the primary task, and predicting mechanism of action is a down-stream task, when such information is available.

### 3.2. Self-Supervised Learning

We consider two of the most successful consistency-based SSL methods, DINO (Caron et al., 2021) and BYOL (Grill et al., 2020), which we briefly describe below and in Appendix C. Both methods use a teacher-student setup to learn a useful feature extractor. A feature extractor is a network intended for other down-stream tasks. The teacher and student network are presented different augmented views of the same image. The teacher network $f_\theta^T$ does not back-propagate any learning signal. It gets updated using the exponential

moving average of the student's weights after each iteration. The student network $f_\theta^S$ is trained to mimic the teacher network, when feed different views of the same image.

In DINO, the student and the teacher share the same architecture – a feature extractor followed by a Multi-Layer Perceptron (MLP) projection head. Given two views of the same image, $x$ and $x'$, the model $f_\theta^S$ is trained to minimize the cross-entropy loss $\mathcal{L}_{\text{DINO}}$ of the output probability of the student $P_S = \text{softmax}(f_\theta^S(x))$ and the teacher $P_T = \text{softmax}(f_\theta^T(x') - C)$. $C$ is a centering vector, defined as the exponential moving average of the teacher's mean pre-softmax activations over each mini-batch. BYOL has an asymmetrical architecture. It shares the same architecture with DINO but the student has an additional MLP network on top of the projection head. The objective is to minimize feature distance between $f_\theta^S(x)$ and $f_\theta^T(x')$. BYOL utilizes two augmented views of the same image whereas DINO uses eight views – two large and six small crops per image.

### 3.3. Cross-Domain Consistency Training

Our method, Cross-Domain Consistency Training (CDCL) is a modification that can be readily applied to common SSL frameworks such as DINO and BYOL described above. CDCL is designed to utilize readily-available metadata (*e.g.* the batch ID the sample was sourced from) to enforce the learning of consistent representations across batches during self-supervision (see Figure 1 and Appendix F).

Consistency-based SSL methods learn by enforcing consistent representations given different augmentations of the same image. In a similar spirit, CDCL enforces consistent representations given a pair of images with the same treatment, but from different batches. When applied to DINO, CDCL creates 1 global crop and 3 local crops from two images sampled from the same treatment but *from different batches* (see Figure 1). When applied to BYOL, CDCL augments two images sampled from the same treatment but from different batches. Then, these views are used as inputs to the teacher and the student network, as described in Section 3.2. Specifically for DINO, we introduce two additional changes, described below, that improve performance even further:

1. Batch-wise centering. We use a centering vector for each batch to mitigate feature collapse.

2. An additional consistency-based loss. We add an extra loss function that increases representational diversity and improves convergence, even for small mini-batches.

These improvements are described below.

**Batch-wise centering**  In DINO, the center vector $C$ is continuously updated using an exponential moving averaging when training. This mechanism is meant to stop the network from representational collapse *i.e.* collapsing to trivial solutions. In CDCL, we use a similar approach but, critically, maintain a center vector $C_d$ for each of the domains $D$. This is done such that the sample probability output of the teacher $P_T$ is dependent on the domain $d$ the input image $x'$ comes from $P_T = \text{softmax}(f_\theta^S(x') - C_d)$. A domain can be considered a batch or a plate, depending on the need. The motivation behind this choice is to ensure that each batch/plate/domain is evenly distributed across the latent space.

**Additional Consistency Loss**   Self-supervised methods like DINO use large mini-batches, but this is computationally demanding. In (Zbontar et al., 2021), the Barlow loss was introduced to reduce the feature representational redundancy and the need for large mini-batches. This loss minimizes the mean square error between the cross-correlation matrix of $P_T$ and $P_S$, of the teacher and the student respectively, and the unit matrix $I$. This loss penalizes cross-feature resemblance, forcing each output dimension to represent a unique uncorrelated feature while enforcing the similarity between $f_\theta^S(x)$ and $f_\theta^T(x')$. We add the Barlow loss as additional loss to reduce the GPU requirements and increase the representational diversity. The combined loss is then defined as $\mathcal{L} = \lambda \mathcal{L}_{\text{DINO}} + (1 - \lambda) \mathcal{L}_{\text{Barlow}}$ regulated by a constant $\lambda \in [0, 1]$. Throughout this work we use $\lambda = 0.25$.

### 3.4. Datasets

We use two different HCI datasets for our experiments, both using setups similar to or the same as the standardized Cell Painting Assay (Bray et al., 2016), which relies on multiplexed fluorescence microscopy data to capture phenotypic response data of cell lines treated with either small molecules or siRNAs. We use RXRX1-HUVEC (Taylor et al., 2019) as our primary dataset consisting of 24 distinct batches and 1139 unique siRNA treatments. We run additional experiments on a subset of CPG0004 (Way et al., 2022) that includes 570 unique treatments with known mechanism-of-action (MoA) data, describing which molecular target a treatment affect, we use this as a downstream task. We down-sample all images to $256 \times 256$ and we split them at the batch-level to create the training, validation and test sets. Additional information can be found in Appendix A.

## 4. Experiments

Our experiments explore the use of self-supervised representational learning for high content screening data. We begin by establishing supervised baselines, and then show the unexpected failure of SSL methods. We pinpoint the source of the problem to batch effects, and demonstrate that CDCL can mitigate batch effects and improves performance, making self-supervision feasible on HCS data. Further studies suggest that our method learns more robust and distinguishing features that are more useful for downstream analysis.

For all the experiments we use DEIT-S (Touvron et al., 2021) models pretrained on IMAGENET (Deng et al., 2009). Inline with previous work, we found that pre-trained ViT-based models outperform similar capacity CNN-based architectures when applied to HCS data (Matsoukas et al., 2021, 2022). For each experiment we report the mean and variance using $k$-fold validation. For the SSL runs, we follow the $k$-NN evaluation protocol, as in (Caron et al., 2021), and the linear evaluation where we train a linear layer on the model's extracted features. To visualise the feature-space we utilize UMAP (McInnes et al., 2018) and we use the k-BET-score (Büttner et al., 2019) to measure sample-mixing between batches. We use Z-norm whitening for batch-wise feature normalization as standard post-processing to counteract batch effects. Further implementation and evaluation details can be found in Appendix B.

Table 1: *Main results* Cross-validation accuracy and k-BET scores on RXRX1-HUVEC.

| Method | Lin. Accuracy ↑ | $k$-NN Accuracy ↑ | Z-norm $k$-NN Accuracy ↑ | k-BET ↑ |
|---|---|---|---|---|
| Supervised | **52.56** $\pm$ 5.08 | 51.44 $\pm$ 5.47 | 53.52 $\pm$ 5.53 | 24.05 $\pm$ 17.76 |
| SSL-DINO | 14.46 $\pm$ 2.65 | 14.28 $\pm$ 2.04 | 25.04 $\pm$ 3.29 | 6.87 $\pm$ 9.45 |
| SSL-BYOL | 4.04 $\pm$ 0.86 | 9.38 $\pm$ 1.45 | 15.1 $\pm$ 2.74 | 17.00 $\pm$ 13.72 |
| SSL-DINO-CB | 30.08 $\pm$ 8.21 | 41.52 $\pm$ 8.5 | 55.3 $\pm$ 7.29 | 12.54 $\pm$ 9.19 |
| SSL-BYOL-CB | 46.24 $\pm$ 4.41 | 42.06 $\pm$ 4.19 | 47.44 $\pm$ 4.02 | **35.73** $\pm$ 17.52 |
| CDCL | 49.64 $\pm$ 4.93 | **53.56** $\pm$ 6.61 | **63.1** $\pm$ 6.33 | 22.26 $\pm$ 14.50 |

Supervised      Dino      CDCL      CDCL + Z-whitening

Figure 2: *Feature embeddings.* UMAP of the test set feature representation for each method, colored by batch (zoomed area is colored by treatment). Dino shows clear batch-wise clustering, indicating that the feature space is dominated by undesirable batch effects, while the zoomed region displays a mixture of treatments. On the other hand, when CDCL + Z-whitening is used the batch effects are, almost, indistinguishable and the zoomed area shows clear treatment-wise clustering.

## 4.1. Self-Supervised Learning

We start by asking: can self-supervised methods like Dino and BYOL learn meaningful representations for HCS image data? Given the relatively large size of high content screening datasets – and their sparse and noisy labels – one would expect these methods would work particularly well in this setting. Through a series of experiments, we explore how poorly off-the-shelf self-supervised methods Dino and BYOL perform on high content data.

The results for RXRX1-HUVEC appear in Table 1, where we compare a standard "weakly"-supervised learning approach (top row) with self-supervision using Dino (SSL-Dino) and BYOL (SSL-BYOL). The performance using self-supervision drops precipitously across all three metrics when compared to the standard supervised approach. This stands in stark contrast to the natural domain, where SSL methods perform on par with their supervised counterparts (Caron et al., 2021).

To understand the root of the problem, we visualise how the extracted features are organised in Figure 2. Each color represents the batch the data was created in (the treatments in the zoomed-in area). Evidently, SSL-learned features are clustered based on the batch they belong to, instead of biologically relevant properties, revealing the dominance of batch effects. Using k-BET (Table 1 and Table 4), we measure how well the learned features are mixed, with respect to the batch they belong to. We find that features are dominated by

Table 2: *Additional* results. **a)** 1-NN Not-Same-Compound MoA accuracy in CPG0004. **b)** Performance impact when training using different data subsets.

| Pre-trained method | Finetuned | 1-NN MoA Acc ↑ |
|---|---|---|
| IMAGENET | ✗ | 8.3 ± 0.7 |
| Weakly Supervised | ✗ | 11.3 ± 2.1 |
| CDCL | ✗ | **14.3** ± 1.8 |
| IMAGENET | ✓ | 21.8 ± 4.0 |
| Weakly Supervised | ✓ | 21.0 ± 5.0 |
| CDCL | ✓ | **22.6** ± 4.5 |

**a)** CPG0004 *results*

| Setting | Method | Norm $k$-NN Accuracy ↑ |
|---|---|---|
| I | Supervised | 18.5 ± 1.89 |
| | CDCL | **20.8** ± 2.15 |
| II | Supervised | 51.68 ± 4.41 |
| | CDCL | **61.43** ± 5.03 |
| III | Supervised | 42.2 ± 3.57 |
| | CDCL | **54.47** ± 5.39 |

**b)** *Exploratory data settings*

batch effects, especially in the training set (see Table 4). This explains the unexpected failure of SSL methods in Table 1. One way to counteract batch effect is Z-norm whitening. Results from Table 1, column 3, show that both DINO and BYOL's k-NN performance is improved, however, this is not enough to combat the problem.

## 4.2. Cross-Domain Consistency Learning

In Section 3, we outlined simple changes to SSL methods that should mitigate the issues caused by batch effects. Below, we explore the effect of this strategy in DINO and BYOL and in Appendix F we provide an expanded explanation of the intuition of our approach.

When cross-batch (CB) learning is applied, denoted as (SSL-DINO-CB and SSL-BYOL-CB), we observe a significant increase in performance of self-supervised methods (see Table 1). When z-norm whitening is applied, the accuracy further improves by a large margin. The feature maps visualized in Figure 2 supported by k-BET scores from Table 1 indicate an obviously decreased feature dependence of the batch effects. Evidently, cross-batch learning helps to learn features that cluster based on their biological effects and reduce dependence on batch effects. When we apply batch-wise centering along with the additional consistency loss, as described in Section 3.3, we find that DINO significantly outperforms the supervised version – even without fine-tuning. CDCL–DINO results in well-behaved features, alleviated from batch effects, and pushes the score by +9.6% over its supervised counterpart.

**CDCL improves downstream performance** We demonstrated that CDCL produces well behaved features that improve accuracy and mitigate batch effects. But are these features useful for downsteam tasks? Is there an advantage of CDCL pre-training? We answer these questions using CPG0004, where predicting the Mechanism of Action (MoA) is the down-stream task. We report the results in Table 2a. Without fine-tuning using MoA labels, CDCL outperforms its weakly supervised counterpart by 3%, demonstrating the versatility of the features learned with CDCL. When fine-tuned, models trained with CDCL outperform all others, including the fully supervised model.

**CDCL increases robustness in exploratory data settings** One of the main purposes of high content screens is to explore the compound space for promising new candidate drugs. This often entails the use of data coming from unseen treatments and batches. To assess performance in different exploratory settings, we subsample RXRX1-HUVEC and we simulate three different scenarios: *I)* many samples per class – using only the controls, *II)* few samples per class – using only the treatments and, *III)* using 50% of the treatments.

We compare CDCL with the weakly supervised baselines and we report our findings in Table 2b. For all cases, we observe a performance drop, compared to the full dataset. However, CDCL consistently outperforms its weakly supervised counterpart. Particularly when only a half of the treatments are available, suggesting that the method is capable of generating features that are unique and therefore more useful for distinguishing new unseen treatments.

## 5. Conclusion

In this work we show that self-supervised learning for high content screens is extremely susceptible to batch effects, and fails to show the same promising results observed in the natural domain. We identify the root causes of the issue and propose methods to counteract them. We demonstrate that using CDCL, metadata-guided SSL can not only match but exceed supervised learning performance in the HCS drug discovery setting. We also show that it is more robust when used on non-ideal exploratory datasets. This is a step towards more domain-agnostic features in biomedical image analysis. The extent to which this is applicable to medical and natural data is an open question which we leave as feature work.

## Acknowledgments

This work was supported by the Wallenberg Autonomous Systems and Software Program (WASP), Stockholm County (HMT 20200958), and the Swedish Research Council (VR) 2017-04609. We acknowledge the use of Berzelius computational resources provided by the Knut and Alice Wallenberg Foundation at the National Supercomputer Centre. We thank Riku Turkki for the thoughtful discussions.

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

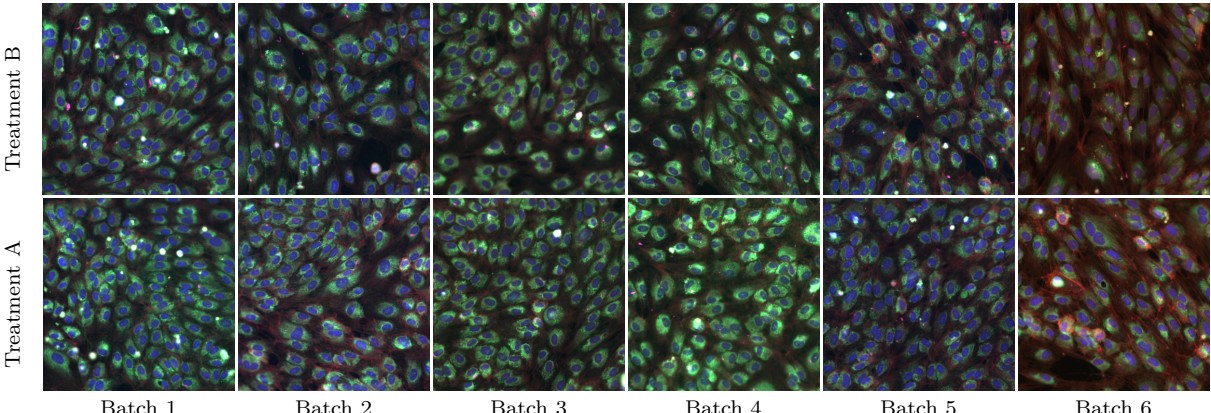

Figure 3: *Example images from* RXRX1-HUVEC showing two different treatments (columns) in 6 different batches (rows).

**Appendix overview** We provide further details and results from the methods and experiments carried out in this work. Starting with Section A, we describe the datasets we use and in Section B we provide additional implementation details. Section C further explains the SSL methods we utilise (Dino (Caron et al., 2021) and BYOL (Grill et al., 2020)) and Section D defines the Barlow loss (Chen and He, 2021), used in CDCL-Dino. In Section E we provide a quantitative evaluation of batch effects and how CDCL helps to mitigate them. Finally, in Section F we portray the intuition behind CDCL and its effectiveness when applied in HCS data.

## Appendix A. Datasets

High content screening allows for standardization and assessment of drug treatments at scale. Assays like Cell Painting in the JUMP-CP initiative (Bray et al., 2016) have generated phenotypic response data of high importance for AI-based drug discovery. In Figure 3 we show image examples from RXRX1-HUVEC, which we use as our primal dataset. We can immediately see that the images are dominated by batch effects, suppressing biological changes due to treatments or other perturbents. In this work, we use two HCS datasets as described in section 3. We describe these datasets more in depth in the section bellow.

**RXRX1-HUVEC** is a subset of the fluorescent microscopy dataset RXRX1 (Taylor et al., 2019), originally designed to study representational learning under batch effects. The subset contains 59,050 6-channel images, consisting of 24 distinct batches *i.e.* replicates of the same experiment. Every experiment contains the same 1139 unique siRNA treatments, which are considered as distinct classes. The task for this dataset is to identify the treatment across batches. To this end we split the data at the batch-level and use 16 batches for training, 4 for validation and 4 for testing, in a 6-fold cross validation setup.

**CPG0004** is a dataset from the Cell Painting Gallery (Way et al., 2022). The full dataset contains over 9412 unique compound-concentration pairs of 5-channel images. We use a well-annotated subset of 570 unique treatments with known Mechanism of Action (MoA) data, where each treatment belongs to one of 50 MoA, coming from (Corsello et al., 2017). The

primary task for this dataset is to predict treatment, the downstream task is to predict MoA, evaluated using the commonly used 1-NN Not-Same-Compound MoA accuracy. CPG0004 consists of 136 plates, with most plate layouts being replicated 5 times. We use a 5-fold cross-validation setup, splitting the data by replicate group and using three for training, one for validation and test respectively.

## Appendix B. Implementation and Evaluation details

**Implementation details** For all the experiments we use DEIT-S models as we found though experimentation that ViT-based models outperform their CNN-based counterparts when IMAGENET pretraining is utilized. Similarly to Matsoukas et al. (2021, 2022) we notice that the benefits of transfer learning from the natural to medical imaging domain increase for models with less inductive bias i.e. DEITs. For establishing the supervised baselines the the DEIT model is followed by a linear layer. For the SSL methods the same model is used as a feature extractor $f_{\theta_{Ext}}$ and we assess performance using the $k$-NN evaluation protocol on the extracted features, similarly to (Caron et al., 2021). All models are trained using AdamW (Loshchilov and Hutter, 2017) as the optimizer. Hyper-parameters are selected through grid search based on results from the validation set. We use learning rate warm-up for 1000 iterations followed by cosine annealing.

The SSL-DINO and SSL-BYOL method were trained following the same protocol as described in their respective papers (Caron et al., 2021; Grill et al., 2020), with minor changes to the augmentation pipeline to make them applicable to the 5 and 6 channel datasets used in this manuscript. The SSL-DINO-CB and SSL-BYOL-CB using the Cross-Batch sampling technique suggested in the manuscript, used the same settings as SSL-DINO and SSl-BYOL except for the sampling technique, which is described in section 3.3. Similarly, CDCL use the same settings as SSL-DINO-CB except for the addition of the *Batch-Wise Centering* and *Additional Consistency Loss* which were added as described in 3.3 The exact implementation details for each setting can be found in the github repository associated with manuscript upon publication.

**Evaluation protocol** We evaluate model performance using three different strategies depending on the type of training. For supervised models, we append a linear layer on top of the feature extractor and we fine-tune the full model. For self-supervised learning, we use the model's output features to make predictions using the following methods: *i)* by adding a linear layer that is learned independently of the feature extractor, which does not back-propagate any learning signal to the feature extractor). *ii)* using the $k$-NN evaluation protocol. The performance is evaluated in terms of treatment-class accuracy for weakly supervision and MoA for full supervision. We visualize the feature space based on the models output features using UMAP (McInnes et al., 2018) in Figures 1 and 2. In both the k-NN evaluation and UMAP feature visualisation we also perform feature normalization as a post processing step as commonly done in HCS datasets. We use Z-norm whitening for batch wise feature normalization, such that each batch has zero mean and unit variance, based on the controls. For all samples $X_b$ belonging to batch $b$, the control samples are used to calculate mean $\mu_b$ and standard deviation $\sigma_b$ and the normalized features $Z_b$ calculated as follows $Z_b = \frac{X_b - \mu_b}{\sigma_b}$. More advanced post-processing methods were also explored, like

TVN (Ando et al., 2017) and COMBAT (Johnson et al., 2007), but no clear performance improvements were observed.

We further include the k-BET-score (Büttner et al., 2019), to measure sample mixing between batches. This metric ranges from 0 to 1, with a higher value indicating more overlap between samples from different batches. Grit score was also used to evaluate the consistency of each compound representation across replicates, using the *cytominer* package (Singh et al., 2020).

## Appendix C. Self-Supervised Learning with Dino and BYOL

As described in Section 3.2, we consider two of the most successful consistency-based SSL methods, DINO (Caron et al., 2021) and BYOL (Grill et al., 2020). Here, we provide additional information about these methods. The goal of these SSL methods is to train a feature extractor that can be used for down-stream tasks. The feature extractor is trained in a teacher-student setup. Both methods are designed to minimize the representational discrepancy between two different augmented views of the same image. In practice this means that the teacher $f_\theta^T$ and student $f_\theta^S$ networks are each given an augmented view of the same image and the representational discrepancy at the output is minimized. This loss signal is only back propagated through the student, and the teacher is updated using the exponential moving average of the student's weights. The teacher $f_\theta^T$ and the student $f_\theta^S$ networks have two primary components, a feature extractor a the projection head. The feature extractor is usually implemented using one of the mainstream architectures *e.g.* DEIT (Touvron et al., 2021) or RESNET (He et al., 2016) and the projection head is an MLP.

In DINO, the student $f_\theta^S = f_{\theta_{Proj}}^S \circ f_{\theta_{Ext}}^S$ and the teacher $f_\theta^T = f_{\theta_{Proj}}^T \circ f_{\theta_{Ext}}^T$ share the same architecture – a feature extractor $f_{\theta_{Ext}}$ followed by a 3-layer MLP layer $f_{\theta_{Proj}}$. Given two views of the same image, $x$ and $x'$, the model $f_\theta^S$ is trained to minimize the cross-entropy loss of the output probability of the student $P_S = softmax(f_\theta^S(x))$ and the teacher $P_T = softmax(f_\theta^T(x') - C)$. $C$ is a centering vector, defined as the exponential moving average of the teacher's mean pre-softmax activations over each mini-batch. BYOL has an asymmetrical architecture. Similarly to DINO, both the teacher and the student consist of a feature extractor $f_{\theta_{Ext}}$, followed by a projection MLP-based head $f_{\theta_{Proj}}$. However, the student has an additional MLP network $f_{\theta_{Pred}}$ on top of the projection head. The objective is to minimize the negative cosine similarity loss between the outputs of the student $f_\theta^S(x) = f_{\theta_{Pred}}^S \circ f_{\theta_{Proj}}^S \circ f_{\theta_{Ext}}^S(x)$ and the teacher $f_\theta^T(x') = f_{\theta_{Proj}}^T \circ f_{\theta_{Ext}}^T(x')$. The main difference between the two methods is the type of augmentations they use. BYOL utilizes two augmented views $V = 2$ of the same image whereas, DINO uses eight views $V = 8$ in total. In detail, for each image, DINO generates one large augmented view, called a global crop, and three smaller views, called local crops. All crops are used by the student but only the global views are passed through the teacher. Their losses used for BYOL and DINO are defined as follows:

$$\mathcal{L}_{BYOL} = -\frac{f_\theta^S(x)}{\|f_\theta^S(x)\|_2} \cdot \frac{f_\theta^T(x')}{\|f_\theta^T(x')\|_2} \tag{1}$$

Table 3: *Redundancy reduction results.* Cross-validation accuracy and k-BET scores on the RXRX1-HUVEC when using Barlow Loss for Dino and BYOL.

| Method | Lin. Accuracy ↑ | $k$-NN Accuracy ↑ | Z-norm $k$-NN Accuracy ↑ | k-BET ↑ |
|---|---|---|---|---|
| SSL-BYOL-BL | $3.76 \pm 0.55$ | $6.58 \pm 0.59$ | $8.88 \pm 0.52$ | $19.76 \pm 15.74$ |
| SSL-DINO-BL | $15.36 \pm 3.82$ | $15.92 \pm 3.60$ | $21.20 \pm 4.03$ | $11.18 \pm 8.04$ |

$$\mathcal{L}_{\text{Dino}} = \sum_{x \in \{x_1^g, x_2^g\}} \sum_{x' \in \{V\} x' \neq x} -P_T(x) log(P_S(x')) \tag{2}$$

## Appendix D. The Barlow Loss

A drawback of many recent SSL methods such as Dino (Caron et al., 2021), BYOL (Grill et al., 2020), SimCLR (Chen et al., 2020) is the need for large mini-batches to learn good features for downstream tasks. To this end, Zbontar et al. (2021) introduced the Barlow loss, which is intended to reduce the required mini-batch size while still learning good features. The reduction in required mini-batch size effectively reduces the GPU memory requirements necessary to train the model.

The Barlow loss calculates the cross-correlation matrix $R$ between the model's outputs of two different augmented views of the same image, and minimizes the mean square error between the cross-correlation matrix $R$ and the identity matrix $I$. See Equation (3) below. This loss penalizes cross-feature resemblance, and enforces the similarity between $f_\theta^S(x)$ and $f_\theta^T(x')$. Thus, it effectively reduces the representational redundancy as the correlation between two outputs dimensions is penalized, forcing the network to learn unique features for each output dimension. We add this additional loss signal in CDCL for Dino, calculated over the indices $i$ and $j$ of the networks outputs, which is defined as

$$\mathcal{L}_{Barlow} = \sum_i (1 - R_{ii})^2 + \alpha \sum_i \sum_{j \neq i} R_{ij}^2 \tag{3}$$

We investigate the impact of replacing the Dino and BYOL losses with the Barlow loss. We report the results in Table 3. Evidently, replacing the loss functions of DINO and BYOL with the Barlow loss does not offer any benefits. Both models still perform poorly. However, once combined with cross-batch sampling, the Barlow loss brings significant performance boosts when comparing CDCL to SSL-DINO-CB (see Table 1), indicating that the Barlow loss significantly contributes to increased performance, but not in isolation.

## Appendix E. Evaluation of sample-mixing between batches

Learning useful representations of HCS images requires models that are able to distinguish biologically-relevant features, such as phenotypic changes due to treatments. However, in this work, we show that current SSL methods learn biologically uninteresting features caused by domain shifts, known as *batch effects*. To qualitatively assess the result that batch effects have on the feature-space we visualise the embedded space using UMAP (McInnes et al.,

Table 4: *k-BET* We report the average cross-validation k-BET score on the RXRX1-HUVEC, train and test sets.

| Method | Train Set k-BET ↑ | Test Set k-BET ↑ | Test Set Z-norm k-BET ↑ |
|---|---|---|---|
| Supervised | $15.78 \pm 4.35$ | $24.05 \pm 17.76$ | $38.20 \pm 25.20$ |
| SSL-DINO | $00.00 \pm 00.00$ | $6.87 \pm 9.45$ | $27.71 \pm 23.76$ |
| SSL-BYOL | $00.02 \pm 00.03$ | $17.00 \pm 13.72$ | $57.90 \pm 20.49$ |
| SSL-DINO-CB | $2.31 \pm 1.29$ | $12.54 \pm 9.19$ | $45.06 \pm 26.60$ |
| SSL-BYOL-CB | $35.39 \pm 10.32$ | $35.73 \pm 17.52$ | $52.12 \pm 28.04$ |
| CDCL | $10.45 \pm 3.59$ | $22.26 \pm 14.50$ | $52.89 \pm 31.20$ |

2018) in Figure 2. We further investigate this quantitatively using two metrics, $k$-BET and grit score.

### E.1. Evaluation metrics

**$k$-BET score** To assess the extent to which samples from different batches are mixed we use the $k$-BET score (Büttner et al., 2019). Given a sample, $k$-BET considers the distribution of its $k$ closest neighbours and compares it with the global distribution – taken over all samples. Using a $\chi^2$ significance test, if the local distribution is not statistically different from the global one, then the neighbourhood is considered well-mixed and the sample is assigned a score of 1. Otherwise a score of 0 is assigned. The final score is calculated as the mean value of all samples and it represents the percentage of samples that belong in well-mixed neighborhoods. In this work, we set $k$ equal to the 0.5% of all data. While this metric is appropriate to evaluate how well the batches are mixed, it does not consider the unique treatments of each sample. Thus, a high $k$-BET score does not imply distinguishable and consistent features. Only well-mixed.

**Grit score** The grit score (Singh et al., 2020) evaluates how well the replicates of each treatment are clustered and how separable they are from their negative controls. This, effectively assess how distinguishable and consistent the representations are in HCS datasets. One drawback of this metric is that it only assess how well each treatment can be separated from the negative control, not from all other treatments. Meaning that if all treatments are well separated from the negative control but not from all other treatments, the grit score can still be high.

### E.2. The effect of CDCL in sample-mixing

Assessing sample-mixing using a single score can be misleading. To evaluate the uniqueness and consistency of treatment replicates, and their mixing, we use accuracy, k-BET and grit score in combination. Below we discuss the impact that SSL and CDCL methods have with respect to the batch effects and the mixing of treatments during training and inference.

We start with with the feature representation – post training – of the training set. In Table 1, we see that both SSL methods perform significantly worse than the other methods, in terms of accuracy. The k-BET score enhances this picture. As seen in Table 4 (first

Table 5: *grit-score (GS)* We report the average cross-validation grit score on the RXRX1-HUVEC test sets.

| Method | Train set GS ↑ | Test set GS ↑ | Test Set - Z-norm GS ↑ |
|---|---|---|---|
| Supervised | $8.37 \pm 0.38$ | $5.29 \pm 1.05$ | $5.60 \pm 1.07$ |
| SSL-DINO | $2.81 \pm 0.14$ | $2.91 \pm 0.87$ | $3.55 \pm 0.69$ |
| SSL-BYOL | $1.19 \pm 0.30$ | $1.51 \pm 0.71$ | $2.55 \pm 0.56$ |
| SSL-DINO-CB | $4.09 \pm 0.22$ | $3.64 \pm 0.88$ | $4.96 \pm 0.96$ |
| SSL-BYOL-CB | $18.04 \pm 8.45$ | $10.04 \pm 5.98$ | $8.40 \pm 2.36$ |
| CDCL | $6.34 \pm 0.30$ | $4.96 \pm 1.05$ | $6.14 \pm 0.90$ |

column), both SSL-DINO and SSL-BYOL have an average k-BET score of 0.0. This implies that the learning algorithms have managed to perfectly separate the different batches of the training set – which is the main cause of failure for off-the-shelf SSL methods. When CDCL is utilised, the representational space of the training-set is no longer separated by batches to the same extend. Instead, both SSL-DINO-CB and SSL-BYOL-CB reach non-zero k-BET scores, suggesting that the learning is no longer focusing solely on batch effects. Interestingly, SSL-BYOL-CB outperforms all others, in terms of sample-mixing. The grit score follows a similar pattern to the one observed for $k$-BET for the training set.

Assessing the mixing of samples on the test-set corroborates that SSL methods leads to uninteresting features for HCS data. As seen in Table 1, CDCL with normalized features (third column) is the top performing model in terms of accuracy. Similarly, the normalized features of CDCL also reach a high k-BET score of 52.89, as seen in the last column of Table 4. SSL-BYOL-CD follows a similar pattern – high accuracy, accompanied with high $k$-BET and grit scores. Surprisingly, SSL-BYOL and SSL-DINO exhibit non-zero $k$-BET scores, whilst the normalized features of SSL-BYOL reach the highest reported $k$-BET. However, when assessing the test-set performance of SSL methods, including the accuracy and grit score, it becomes clear that the increased $k$-BET score does not imply biologically-relevant features. The samples appear well-mixed but, the low accuracy and grit scores imply that the features are not useful for distinguishing treatments. Note that a high $k$-BET score can be achieved by simply assigning random feature vectors to each sample. Overall, CDCL shows high performance across all three different metrics used, showing that it generates unique and versatile representations that are less impacted by batch effects.

## Appendix F. Why CDCL works for HCS data?

In Section 3, we outline simple changes to SSL methods that should mitigate the issues caused by batch effects. Instead of presenting augmented pairs of the same image to the network during learning, we propose to use readily available metadata to select pairs of samples that have the same treatments but belong to different batches. Results from Section 4 corroborate that CDCL drastically reduces the impact of batch effects. *But why – what makes CDCL to work so well for HCS data?*

The intuition for this approach comes from the guiding principle of self-supervised learning – that the network should learn common features and ignore sources of noise. In HCS

data, each image should in principle contain two dominating signals. The first is related to the treatment and its biological effect. The other is comprised of confounding factors, *i.e.* batch effects. By providing two images of the same treatment but different batches to the network, the common signal should only be the one of interest, the biological. The sampling strategy is not tied to the treatment information. It can be done using any metadata that hint similar biological processes but come from a different experiment. In this work, we use treatment-based sampling for consistency and fair comparison.

