# OpenReview forum: "Metadata-guided Consistency Learning for High Content Images"
_MIDL.io/2023/Conference — MIDL 2023 Poster_

### Official Review · Reviewer_pRsA · 2023-01-29

**Confidence:** 3
**Preliminary Rating:** 3

**Summary:**

This paper revealed that the recent self-supervised learning techniques underperform on high content imaging assays, due to the domain shifts in the data. Thus, the authors introduced a Cross-Domain Consistency Learning (CDCL) method to tackle the problem of batch effects. CDCL enforces the learning of biological similarities while disregarding undesirable batch-specific signals, which leads to more useful and versatile representations.

**Strengths:**

A relatively new and interesting problem. Such problem may be important since the shift will violate some assumptions in typical SSL, leading to inferior performance.
The method may give a perspective in domain-agnostic feature learning, which may benefit other tasks.


**Weaknesses:**

The technical innovation and contribution are still ambiguous. Please clearly highlight the technical innovations. On page 5, it seems that the authors just briefly introduce some small modifications, yet, the motivation of why you applied these techniques is still unclear. Could authors give a more detailed and solid explanation of why they want to introduce the Barlow loss (i.e., why this can reduce the GPU requirements and increase the representational diversity) and what is the advantage of ensuring that each batch/plate/domain is evenly distributed across the latent space?

Authors could provide more intuitive illustrations of what their metadata are and indicate how they use these data in the learning framework. Authors can also provide some background about this metadata in this task.


**Deanonymize Review:**

no

**Paper Type:**

methodological development

**Questions To Address In The Rebuttal:**

The technical innovation and contribution are still ambiguous. Please clearly highlight the technical innovations. On page 5, it seems that the authors just briefly introduce some small modifications, yet, the motivation of why you applied these techniques is still unclear. Could authors give a more detailed and solid explanation of why they want to introduce the Barlow loss (i.e., why this can reduce the GPU requirements and increase the representational diversity) and what is the advantage of ensuring that each batch/plate/domain is evenly distributed across the latent space?

Authors could provide more intuitive illustrations of what their metadata are and indicate how they use these data in the learning framework. Authors can also provide some background about this metadata in this task.

---

### Official Review · Reviewer_13Dw · 2023-02-06

**Confidence:** 4
**Preliminary Rating:** 4
**Recommendation:** Poster

**Summary:**

The authors propose a self-supervised pipeline to extract features from high-content image-based drug screening. In this work, they focused on overcoming the experiment-wise signal variability that normally prevents such methods from finding the relevant features related to the studied treatments. In particular, the combination of cross-domain consistency learning (CDCL) with a self-supervised learning (SSL) method (i.e., DINO) is suggested. They show that their approach is able generalise despite the experimental bias present in the dataset.

**Strengths:**

The manuscript is well-written and organised. All the methods are explained in detail and the authors confirm that the code will be available with the publication of the work, thus promoting the reusability of the approach.
Besides developing a method that overcomes the challenge of normalising the data domain across experimental replicas and groups, the authors compare its performance with state-of-the-art techniques and study when and how these techniques (do not) perform well.
The method is also tested on two different publicly available datasets.


**Weaknesses:**

Probably it is due to the lack of space but the authors provide little detail about the training parameters and how each of the methods should be configurated to obtain similar results.
The authors say that an ablation study was performed when testing different data settings (table 2.b). This type of analysis, however, is not an ablation study of the method's parameters but an analysis of how data distribution affects the outcome of the method.
Likewise, using dataset II in table 2.b, seems that CDCL obtains better results than in the main table 1. How would the authors explain this?  Moreover, for the ablation study, the authors say: "CDCLincreases robustness in exploratory data settings. One of the main purposes of high content screens is to explore the compound space for promising new candidate drugs.". It is not clear from the text and the experiments performed how the information in table 2.b can be used to assess this.

**Deanonymize Review:**

yes

**Detailed Comments:**

The proposed CDCL is particularly interesting when the signal variability is not fully controlled but still, normalisation is very much needed for the method to be able to generalise. This is also a common situation in H&E modality, for instance. In this sense, I have two questions/curiosities about the proposed work:
- If someone wanted to use the trained model / features with new data (for comparison or to identify some treatment for example), how sensitive would be this approach to new signal variability?
- How difficult would it be to integrate CDCL or a similar approach as the preprocessing strategy of any ML-based method? Seems that CDCL manages to "centralise" the distribution of the signal by learning different centres. Such an approach could be beneficial to the analysis of other microscopy modalities.

**Paper Type:**

methodological development

**Questions To Address In The Rebuttal:**

How was the number of centres estimated in CDCL? (Figure 1)

Figure 1: As it is now it's difficult to follow when the authors say top, center or left. I would recommend using a), b) and c) instead. the zoomed areas do not seem to coincide with the highlighted ones (squares) and this also happens in figure 2.

Along the text, the authors comment on the benefits of z-norm whitening. Same as they provide a technical description of the methods, I would suggest also describing this one.

Please, define what is MLP with all the words.

Table 1, please describe in the legend what is CB and CDCL.

---

### Official Review · Reviewer_3hxA · 2023-02-09

**Confidence:** 2
**Preliminary Rating:** 4
**Recommendation:** Oral

**Summary:**

Full disclaimer: this is an emergency review, and I forgot the printed paper at the office, so I am running at a handicap.

This paper is about self-supervised classification (clustering?) of high-content images (?), which is used to assess drugs effectiveness. Despite the potential to automate this task, it seems that it remains confronted to the issue (as many applications) of data availability.

The authors propose to refine recent self-supervised methods, DINO and BYOL, that, while working well on natural images, perform rather badly on this medical task (I would assume that the huge variability in natural images makes it for some implicit regularization, which is not the case here). The authors exploit some task-specific prior to regularize things manually, by grouping parameters moving averages by "batch" (in the physical experiment, not DL training).

For now I will stick to a conservative weak accept (as I am quite unsure about myself). But this could be easilty be upgraded during rebuttal, as by then I manage to get a clearer view of the paper. I would warn the authors to be responsive during the rebuttal, to be able to engage in a discussion (item 5 in the [reviewer guidelines](https://2023.midl.io/reviewer-guidelines.html)), and not wait for the deadline to submit a response.

**Strengths:**

- The paper is well written and easy to follow
- The limitation of DINO and BYOL on medical images, seem coherent, and the proposed solution consistent.
- Seemingly very good results

`Additional characters required: 18`

**Weaknesses:**

Due to my review timeline, I did not spot any major weakness (for now). Not that I am looking for some specifically, but this might pop-up during the rebuttal.

Minor: the task could perhaps be described better, for the uninitiated.

**Deanonymize Review:**

no

**Detailed Comments:**

Pedantic notes (for the camera ready):
- write names in math in `\text` mode, such as $\text{softmax}$ and not $softmax$ (by default, each letter is considered as an independent variable, which are in italics and with a slightly wider spacing). Same for $\mathcal L_\text{DINO}$
- I think the authors use too much emphasis at times, this could (should?) be reduced, esp in section 4.1 and 4.2
- Tables captions should be on top of the table, on below. Simply moving `\caption{}` before will do the trick
- Use `\eqref` for equations, not the base `\ref` (it should be "Equation (3)" and not "Equation 3")
- If you are going for `\textsc` for ImageNet, it should be the same for `\textsc{Dino}` (the original authors messed up their French typography class), perhaps also for BYOL (don't know how it is pronounced)
- text of Figure 1 should be slightly bigger for better legibility

**Paper Type:**

methodological development

**Questions To Address In The Rebuttal:**

I see that each "batch" is a repeat of the (same) experiment? What does change then, are those the exact same lab conditions, or is there some variability introduced ? (Temperature, humidity, cosmic rays, ..)

Given the dissimilarity between natural images and the task, does it make really sense to pre-train on imagenet?

What are the images original resolutions? Do they all have the same scale? Isn't the 256×256 resized resolution too small?

I take it that, in Equation (3), $i$ iterates over the mini-batch size, or did I mis-understand?

What is the Cross Batch (CB) in `SSL-DINO-CB` ? I was going to ask, what are the performances of DINO and BYOL with the redundancy loss? Is that it, or is this something else?

---
**Final rating:** I thank the reviewers for their response to my questions. I still feel rather inadequate judging this paper, and I am keeping my rating of `weak accept`, with a confidence of `2`.

---

### Meta-Review · Area_Chair_FEEc · 2023-02-25

**Recommendation:** Accept (Poster)
**Confidence:** 4

**Metareview:**

The authors show that standard self-supervised learning techniques underperform on high content imaging assays, due to domain shifts. They introduced a cross-domain consistency learning method, embedding task-specific priors and leveraging the available metadata. The reviewers agree that this is a relatively new and interesting problem. Also, the paper is well written. During discussion, the authors provided comprehensive answers, including additional results. They provided a good clarification of the technical contribution to Reviewer pRsA.